# Comparison of UKESM1 and CESM2 Simulations Using the Same Multi-Target Stratospheric Aerosol Injection Strategy

Matthew Henry[1], Jim Haywood[1,2], Andy Jones[2], Mohit Dalvi[2], Alice Wells[1], Daniele Visioni[3,4], Ewa M. Bednarz[3,5,6], Douglas G. MacMartin[3], Walker Lee[3], and Mari R. Tye[4,7]

[1]Department of Mathematics, University of Exeter, Exeter, UK
[2]Met Office Hadley Center, Exeter, UK
[3]Sibley School of Mechanical and Aerospace Engineering, Cornell University, Ithaca, NY, USA
[4]National Center for Atmospheric Research, Boulder, CO, USA
[5]Cooperative Institute for Research in Environmental Sciences (CIRES), University of Colorado Boulder, Boulder, CO, USA
[6]NOAA Chemical Sciences Laboratory (NOAA CSL), Boulder, CO, USA
[7]Whiting School of Civil Engineering, Johns Hopkins University, Baltimore, MD, USA

**Correspondence:** Matthew Henry (m.henry@exeter.ac.uk)

**Abstract.** Solar climate intervention using stratospheric aerosol injection (SAI) has been proposed as a method which could offset some of the adverse effects of global warming. The Assessing Responses and Impacts of Solar climate intervention on the Earth system with Stratospheric Aerosol Injection (ARISE-SAI) set of simulations is based on a moderate greenhouse gas emission scenario and employs injection of sulphur dioxide at four off-equatorial locations using a control algorithm which maintains the global-mean surface temperature at 1.5 K above preindustrial conditions (ARISE-SAI-1.5), as well as the latitudinal gradient and inter-hemispheric difference in surface temperature. This is the first comparison between two models (CESM2 and UKESM1) applying the same multi-target SAI strategy. CESM2 is successful in reaching its temperature targets, but UKESM1 has considerable residual Arctic warming. This occurs because the pattern of temperature change in a climate with SAI is determined both by the structure of the climate forcing (mainly greenhouse gases and stratospheric aerosols) and the climate models' feedbacks, the latter of which favour a strong Arctic amplification of warming in UKESM1. Therefore, research constraining the level of future Arctic warming would also inform any hypothetical SAI deployment strategy which aims to maintain the interhemispheric and equator-to-pole near-surface temperature differences. Furthermore, despite broad agreement in the precipitation response in the extratropics, precipitation changes over tropical land show important inter-model differences, even under greenhouse gas forcing only. In general, this ensemble comparison is the first step in comparing policy-relevant scenarios of SAI, and will help in the design of an experimental protocol which both reduces some known negative side effects of SAI and is simple enough to encourage more climate models to participate.

## 1 Introduction

The 2015 Paris agreement's goal was to limit global warming to well below 2 K, preferably below 1.5 K, above pre-industrial levels. This temperature target is seen as a threshold for climate safety (Masson-Delmotte et al., 2018), with temperatures beyond 1.5 K potentially triggering multiple climate tipping points (Armstrong McKay et al., 2022). The realisation by the

scientific community of the difficulty of limiting global mean temperatures to within these 1.5 or 2 K targets through conventional emission reductions of carbon dioxide (Rogelj et al., 2016; Millar et al., 2017; Tollefson, 2018), or short-lived climate forcing agents (e.g Jones et al., 2018) has led to increased calls for research into climate interventions which aim to partially offset global warming by increasing planetary albedo. These are known as solar radiation modification, solar geoengineering, or solar climate interventions. These techniques may be used to stabilise near-surface temperatures while societies cut emissions and remove greenhouse gases from the atmosphere. There is growing support for researching solar climate interventions; for example, the National Academies of Sciences Engineering and Medicine (2021) report recommends an initial investment of \$200 million over five years into solar geoengineering research and proposes ways to effectively govern this research.

One of the most prominent methods of solar climate interventions in the scientific literature is stratospheric aerosol injection (SAI), which involves injecting aerosols or their precursors, in the lower stratosphere. SAI was first proposed by Budyko (1977) and then by Nobel Prize winner Paul Crutzen (Crutzen, 2006) who noted that reductions in tropospheric aerosols in pollution abatement policies may add to the warming caused by greenhouse gases. He also concluded that any detrimental impacts on stratospheric ozone caused by stratospheric aerosols might be a price worth paying in order to significantly ameliorate the impacts of global warming. SAI has subsequently been studied mainly by using coupled global circulation models. Those have some uncertainties, which are relevant to both global warming and stratospheric aerosol injection (e.g. Kravitz and MacMartin, 2020). Additionally, difficulties in comparing model outputs owing to the lack of coordination of modelled scenarios and deployment strategies (e.g. Jones et al., 2010) could confound interpretation of results.

The difficulties in comparing results from uncoordinated modelling studies led to the formation of the Geoengineering Model Intercomparison Project (GeoMIP) (Kravitz et al., 2011, 2013, 2015). The most recent set of GeoMIP simulations of SAI (GeoMIP G6) prescribed the reduction of the net radiative forcing from a high-end forcing scenario (Shared Socioeconomic Pathway 5-8.5, SSP5-8.5) to a medium forcing scenario (SSP2-4.5) using either a reduction in the solar constant (G6solar) or injection of stratospheric aerosols at the equator (G6sulfur) (Kravitz et al., 2015). Multi-model assessments of the side effects of SAI when injecting at the Equator consistently reveal over-cooling of the tropics, under-cooling of polar regions, and changes in tropical precipitation (Visioni et al., 2021; Jones et al., 2022); it appears that these side-effects can be significantly ameliorated by injecting at multiple different latitudes in the stratosphere (e.g. Kravitz et al., 2017).

In earlier work, a controller algorithm was used to determine how much to inject at different locations in the stratosphere (15 and 30 degrees North and South) to maintain the equator-to-pole and inter-hemispheric difference in surface temperature in addition to the global-mean temperature (Kravitz et al., 2017; Tilmes et al., 2018). The baseline scenario was the high-end Representative Concentration Pathway 8.5 (RCP8.5) emission scenario and stratospheric sulphur injection started in 2020. These simulations show that the interhemispheric and equator-to-pole temperature targets can be met, even in a high greenhouse gas emission scenario in CESM1. However, they also show that the hydrological cycle is suppressed relative to the target climate, results that appear common to many other SAI strategies (e.g. Tilmes et al., 2013; Irvine et al., 2019). MacMartin et al. (2022)

and others (Tilmes et al., 2018) subsequently argued for a more plausible set of scenarios to inform policy-makers, with a later start date and lower baseline greenhouse gas emission scenario. Richter et al. (2022) then simulated SAI using the Community Earth System Model version 2, Whole Atmosphere Community Climate Model (CESM2-WACCM) with 10 ensemble members in a moderate baseline emission scenario (SSP2-4.5) to maintain temperatures at 1.5 K above pre-industrial levels using multiple injection locations. In order to replicate this multi-target scenario in multiple climate models, the response to fixed single-point $SO_2$ injections at a range of latitudes was compared in multiple models, including UKESM1 (Visioni et al., 2023a; Bednarz et al., 2023). In this paper, we present a new comparison of SAI simulations with UKESM1 under the same GHG emission scenario and same multi-target, multi-latitude SAI strategy as Richter et al. (2022).

## 2 Methods

### 2.1 Models Description

The first set of simulations considered in this study, first presented in Richter et al. (2022), was conducted using the the National Center for Atmospheric Research (NCAR) Community Earth System Model, version 2 (CESM2) with the Whole Atmosphere Community Climate Model version 6 as its atmospheric component (CESM2(WACCM6)) (Gettelman et al., 2019; Danabasoglu et al., 2020). The atmospheric component (WACCM) has a 1.25° longitude by 0.9° latitude resolution, and 70 vertical levels with a model top at 140km. The tropospheric physics are the same as in the lower top configuration, the Community Atmosphere Model Version 6 (CAM6). CESM2(WACCM6) uses prognostic aerosols represented using the Modal Aerosol Model version 4 (MAM4) (Liu et al., 2016), and also includes a comprehensive chemistry module with interactive tropospheric, stratospheric, mesospheric, and lower thermospheric (TSMLT) chemistry with 228 prognostic chemical species, described in detail in Gettelman et al. (2019). Finally, the ocean model is based on the Parallel Ocean Program version 2 (POP2, Danabasoglu et al. (2020)).

The second set of simulations presented in this article used the U.K. Earth System Model UKESM1.0 (Sellar et al., 2019). UKESM1 consists of the physical atmosphere-land-ocean-sea ice model HadGEM3-GC3.1 (Kuhlbrodt et al., 2018) and uses the Met Office Unified Model (UM) as its atmospheric component, which has a 1.875° longitude by 1.25° latitude resolution, and 85 vertical levels with a model top at  85 km. The model includes the United Kingdom Chemistry and Aerosol (UKCA) chemistry model (Mulcahy et al., 2018; Archibald et al., 2020), with troposphere-stratosphere chemistry and coupling to a multi-species modal aerosol scheme (Mann et al., 2010). For a more complete description of the UKESM1 model configuration used in the SAI scenarios the reader is referred to the GeoMIP study of Jones et al. (2022).

## 2.2 Simulations Description

The reference simulations use the SSP2-4.5 scenario, which follows on from the Representative Concentration Pathway 4.5 (RCP4.5) scenario, and is considered a "middle-of-the-road" intermediate mitigation scenario (O'Neill et al., 2016), which is arguably close to our projected emission trajectory (Pielke Jr et al., 2022). The reference SSP2-4.5 simulations begin in 2015 and run until 2100. The CESM2 ensemble comprises 10 members, and the UKESM1 ensemble comprises 5. The SAI simulations branch from SSP2-4.5 beginning in 2035 and run until 2070.

The stratospheric aerosol injection (SAI) simulations are part of a set of different solar climate intervention (SCI) implementation scenarios collectively called "Assessing Responses and Impacts of Solar climate intervention of the Earth system", or "ARISE", and the SAI simulations are denoted "ARISE-SAI". Following MacMartin et al. (2022) and Richter et al. (2022), we begin SAI at 2035 with the target of maintaining global-mean surface temperatures at 1.5 K above pre-industrial levels. Hence, this simulation set is called ARISE-SAI-1.5, other ARISE-SAI simulations are planned with different temperature targets and start dates. The stratospheric aerosol injection occurs at an altitude of 21.5 km at four locations: 15°N, 15°S, 30°N, and 30°S. The longitude of injection is 180° for both UKESM1 and CESM2. The control algorithm starts with a "best-guess" (or feed-forward) for the desired injection rates, which is then corrected by feedback. In this case, the initial guess is set to only apply injection at 15°N and 15°S to manage the global-mean surface temperature (T0). The control algorithm (MacMartin et al., 2018; Kravitz et al., 2017) then (i) adjusts the total amount of injection across all four latitudes to maintain T0, (ii) adjusts the balance between Northern and Southern Hemisphere injection rates to maintain the hemispheric temperature difference (T1), and then (iii) adjusts the balance between 15°N/S and 30°N/S to maintain the equator-to-pole temperature difference (T2). Importantly, the priority is chosen in that order; as the injection rate cannot be negative at any latitude, this introduces constraints on simultaneously meeting the multiple objectives (Lee et al., 2020). T1 and T2 are defined in equation 1 from Kravitz et al. (2017). The integral and proportional control gains are 0.0183, 0.0753, and 0.3120 for T0, T1, and T2 respectively, for both CESM2 and UKESM1.

The values for the temperature targets T0, T1, and T2 are based on the 2020-2039 mean of the SSP2-4.5 simulations for CESM2 (288.64K, 0.8768K, and -5.89K respectively) (Richter et al., 2022) and on the 2014-2033 mean for UKESM1 (288.06K, 0.54K, -6.05K respectively). The target period for CESM2 was chosen as representative for when the climate might reach 1.5 K (MacMartin et al., 2022); this is not based on the ensemble of the CESM2 SSP2-4.5 scenarios, but on a combination of observational temperature trend extrapolations and estimates from climate models (the median estimate for crossing that threshold from CMIP6 models is 2028, but with a large uncertainty (Tebaldi et al., 2021)). In 2020-39, CESM2 reaches 1.58K above its preindustrial level (287.06K), and it reaches 1.5K above preindustrial in 2016-2035. The target period for UKESM1 was chosen based on the UKESM1 historical simulations as the 20-year period over which the global-mean surface temperature value exceeds UKESM1's preindustrial value by 1.5 K. In what follows, comparisons will be made relative to each model's own reference period.

# 3 Results and Discussion

## 3.1 Temperature Targets

Figure 1 shows the global-mean near-surface air temperature (T0), inter-hemispheric temperature gradient (T1), and the equator-to-pole temperature gradient (T2) for all simulations of the reference scenario (SSP2-4.5 in red) and the ARISE-SAI-1.5 scenario (blue). In CESM2, both the interhemispheric temperature gradient and the equator-to-pole gradient do not change much in the reference SSP2-4.5 scenario runs, hence the difference between ARISE-SAI-1.5 and SSP2-4.5 simulations for these metrics is small. UKESM1 ARISE-SAI-1.5 simulations, on the other hand, only reach the global-mean and interhemispheric temperature gradient targets by around 2045. In addition, while the increase in equator-to-pole temperature gradient is smaller in the ARISE-SAI-1.5 simulations compared to the SSP2-4.5 simulations, it does not reach the relevant T2 target. This was also the case for the Geoengineering Large Ensemble Project using CESM1 (Tilmes et al., 2018), though the discrepancy from the T2 target was smaller.

## 3.2 Injection Rates

In the first simulation which used this multi-target strategy in CESM1(WACCM) (Kravitz et al., 2017), the required injection rates were roughly hemispherically symmetric. Subsequently, there was a land model change in CESM1(WACCM), and more injection was needed in the Northern than Southern hemisphere in order to maintain the interhemispheric temperature difference T1 (Tilmes et al., 2018). However, in CESM2(WACCM), the same control algorithm required more injection in the Southern Hemisphere to satisfy T1 (Tilmes et al., 2020; Richter et al., 2022). There are 3 contributors to this difference (Fasullo and Richter, 2022): the fast cloud-adjustment to $CO_2$ in CESM2 results in decreased cloud cover over the Southern Hemisphere requiring more aerosols from SAI to compensate; the North Atlantic warming hole (which has been linked to a reduction in the Atlantic Meridional Overturning Circulation (AMOC), increased heat transport out of the North Atlantic, and a poleward shift of westerly winds as a response to external forcings (Keil et al., 2020; He et al., 2022)) means the Northern hemisphere needs less aerosols from SAI; and the decrease in tropospheric aerosol pollution in SSP2-4.5 (primarily in the Northern hemisphere) is smaller in CESM2 than in CESM1 (due to a change from RCP8.5 to SSP2-4.5) which results in less need for Northern hemisphere mitigation from SAI. The fast cloud-adjustment to $CO_2$ is expected to be different in other climate models (Smith et al., 2020; Wang et al., 2021). However as shown in Fasullo and Richter (2022), separating the fast adjustment from the surface temperature dependent response requires further idealized experiments, which are beyond the scope of the present study. The baseline scenario (SSP2-4.5) is the same for both models; nonetheless, the response to the same aerosol forcing and short-lived greenhouse gases forcing might be different between the two models (Smith et al., 2020). The North Atlantic warming hole, however, is not present in UKESM1 (figure 4) and is expected to be different for other climate models (further discussed in section 3.4). These differences result in a difference in the distribution of injection rates that are required to satisfy the temperature objectives.

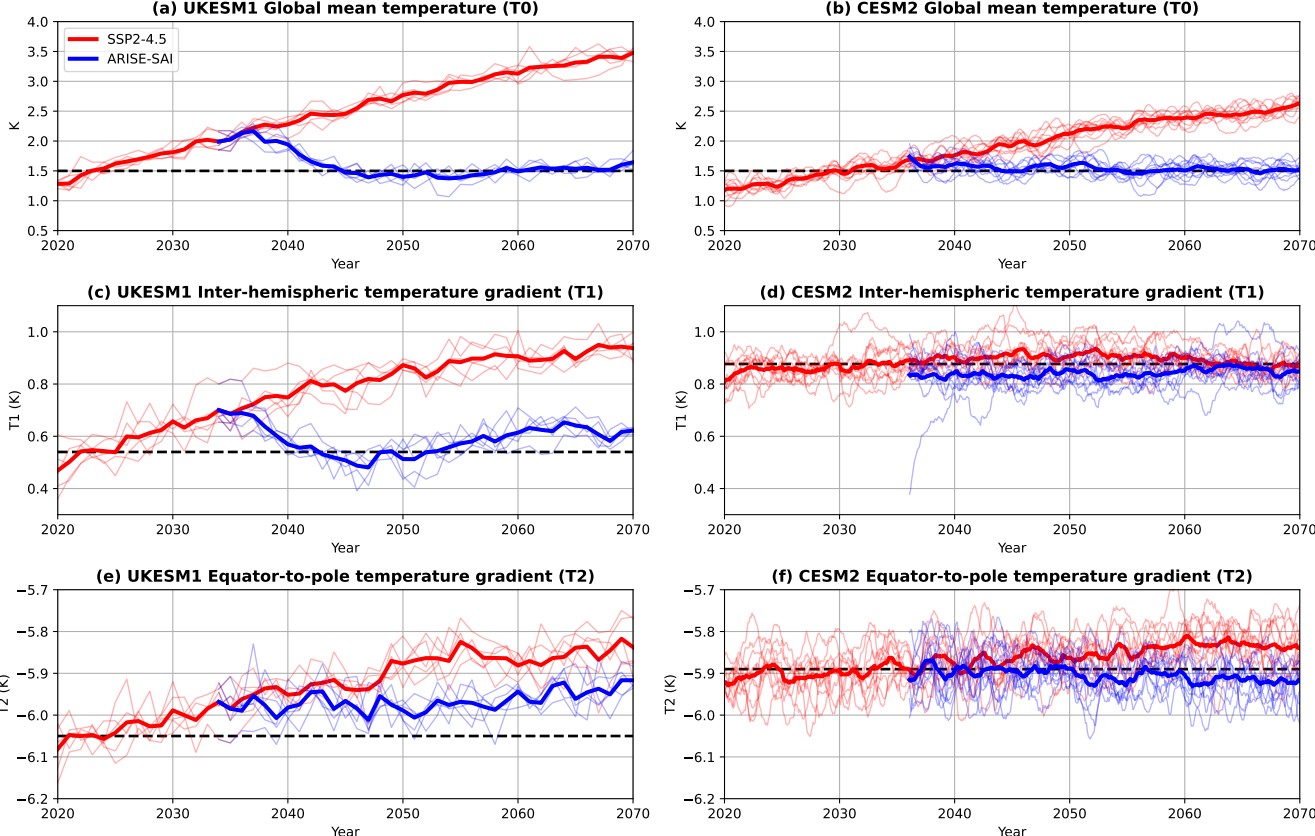

**Figure 1.** Comparison of global-mean temperature (T0), interhemispheric temperature difference (T1), and equator-to-pole temperature gradient (T2) for the SSP2-4.5 simulations (red) and the ARISE-SAI-1.5 simulations (blue) for UKESM1 (left) and CESM2 (right). Thin lines represent individual ensemble members, whereas the thick lines show the ensemble mean. The temperature targets for the controller are shown in the black dashed lines.

A comparison of injection rates between UKESM1 and CESM2 chosen by the controller is given in Figure 2. While CESM2 has a linear increase in injection rates, UKESM1 has a large initial increase followed by a slower increase in the injection rate. This is caused by having the same 2035 start date for both sets of simulations: by 2035, the global-mean temperature in UKESM1 already exceeds its target temperature (defined as the 2014-2033 period), hence it requires a large initial increase in injection. The target period for CESM2 however is 2020-2039, hence by 2035 the temperature has not exceeded its target by much, making the initial increase in injection rates much smoother. Since the initial guess is set to only apply injection at 15°N and 15°S to manage the global-mean surface temperature (T0), the algorithm preferentially injects at those latitudes in the initial decade for UKESM1. Then, the majority of the injection happens at 30°N and 30°S. However, after 2055, there is a marked increase of injection at 15°N while the injections at 30°N and 30°S stabilise. In CESM2, 82% of the injection occurs

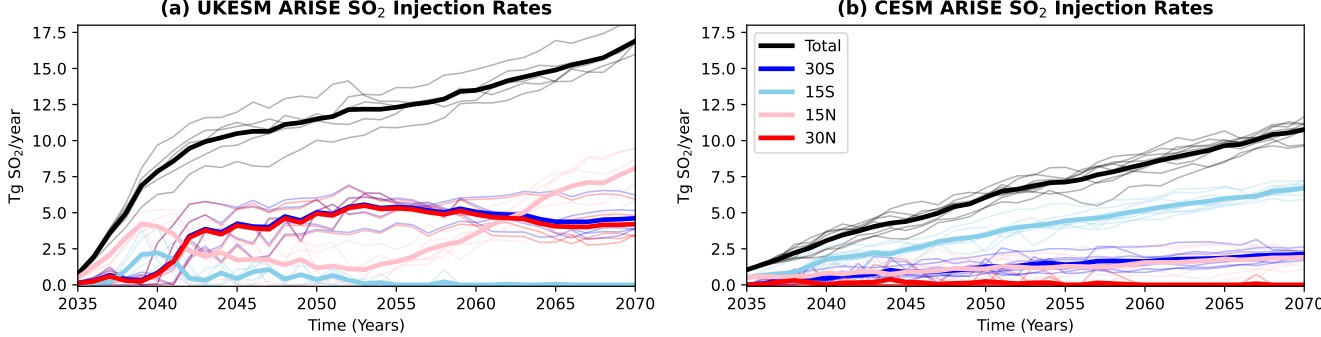

**Figure 2.** Comparison of injection rates at four different latitudes in the stratosphere for UKESM1 and CESM2. Thin lines represent individual ensemble members, whereas the thick lines show the ensemble mean. Note that while CESM2 has 10 ensemble members, UKESM1 only has 5.

in the Southern hemisphere by 2070, whereas in UKESM1 it is only 27%.

In UKESM1, the temperature response to injection at 15°N only and 30°N only are quite similar, in contrast with CESM2
(Visioni et al., 2023a). Thus the influence matrix between injection rates and changes in T0, T1, and T2 is almost singular,
therefore small changes in what the controller needs to respond to can result in large changes in the controller-defined injection
rates, as observed after 2055 in these simulations. In an effort to correct for an increasing error in meeting the T1 objective
caused by continued Arctic warming and slight southern hemisphere cooling after 2050, the controller increases the fraction of
injection at 15N. However, redistributing the injection rates is unable to significantly change T1, resulting in further increases in
the fraction injection at 15N. Therefore the set of injection latitudes chosen in CESM2 during initial studies to simultaneously
meet these temperature objectives (MacMartin et al., 2017) might not necessarily work in other climate models and may need
to be adjusted to ensure that the set of injection latitudes yield distinct influences on the zonal-mean temperature response.
It is worth noting that the decrease in the equator-to-pole temperature gradient from the increase in greenhouse gases is still
moderated by using SAI (figure 1e and 4e,f).

### 3.3 Stratospheric aerosols

Figure 3 compares the increase in stratospheric aerosols in the first (2035-2054, top) and last (2050-2069, bottom) two decades
of the simulations between UKESM1 (left) and CESM2 (right) relative to their respective reference periods. The aerosol distributions are consistent with the injection rates (figure 2). In UKESM1, there is a higher concentration of aerosols in the Northern
Hemisphere in 2050-2069, which is consistent with the increase in injection at 15°N. In CESM2, the aerosols are mostly in the
Southern hemisphere, which is consistent with the majority of the injection occurring at 15°S.

In previous work (Visioni et al., 2023a), the output of single latitude injection simulations was compared. Notably, the sulphate lifetime and increase in $SO_4$ burden are approximately 50% higher for CESM2 than for UKESM1 for a given constant $SO_2$ injection, though this depends somewhat on the latitude of injection. This reduced lifetime is compensated by the total injection being approximately 50% greater in UKESM1 than in CESM2 (figure 2). Hence, the total $SO_4$ burden increase is similar by 2050-69 in the two models (6.2 TgS for UKESM1 and 6.8 TgS for CESM2). Figure 3e and 3f show the increase in aerosol optical depth for both models, the global-mean increases are 0.20 for UKESM1 and 0.12 for CESM2 with 31% of the increase in the Northern hemisphere for CESM2 and 57% for UKESM1. Again, this is consistent with Visioni et al. (2023a), who found that the normalised $SO_4$ increase per 0.1 AOD is approximately 50% larger for CESM2 than for UKESM1. Finally, this agrees with the larger effective radius of aerosols in CESM2 relative to UKESM1 (Visioni et al., 2023a), as for the same mass, smaller particles are generally more reflective hence lead to a higher optical depth.

## 3.4 Surface temperature and precipitation

Figure 4 shows the near-surface air temperature change for SSP2-4.5 and ARISE-SAI-1.5 between the 2050-2069 and the reference periods for UKESM1 and CESM2. Panel e shows the zonal-mean temperature change for both models and scenarios, and panel f shows the zonal-mean surface temperature cooling from SAI compared to the same period of the SSP2-4.5 simulation. The most notable difference between models is the large Arctic amplification in UKESM1 compared to CESM2 (a ratio of 3.6 for UKESM1 compared to 2.1 for CESM2 when defined as warming northward of 70°N relative to the global mean). The coupled nature of the Arctic climate makes it difficult to quantify the role of individual mechanisms in the Arctic amplified warming, which were reviewed in Previdi et al. (2021) and Taylor et al. (2022). The uncertainty in the sea ice feedback would be reduced in a world with SAI as the sea ice would be restored. However, the uncertainties in cloud response to $CO_2$ and aerosol-cloud interactions would be at least as important. The Arctic warming in UKESM1 in the ARISE-SAI-1.5 scenario happens mostly in winter with no warming in summer, this is in contrast to CESM2 which has no seasonality of Arctic temperature change (figure A1). The total Arctic warming in UKESM1 under ARISE-SAI-1.5 is equivalent to the warming of CESM2 under SSP2-4.5 (figure 4e). Moreover, the cooling from SAI is much more Arctic amplified in UKESM1 (figure 4f), even though the T2 target was not reached in that model. The latitudinal pattern of the AOD increase (roughly hemispherically symmetric for UKESM1 and Southern hemisphere amplified for CESM2) is poorly correlated with the pattern of cooling (Arctic amplified for both models). This indicates that the pattern of surface cooling is dominated by the model's climate feedbacks rather than the pattern of the direct radiative forcing from aerosol scattering. This will also have an impact on the distribution of $SO_2$ injection : until the amount of future Arctic warming is further constrained, it will be hard to determine which injection strategy will maintain T1 and T2.

Another notable difference is the North Atlantic warming hole, which is present in CESM2 but not in UKESM1. The North Atlantic warming hole has been associated with oceanic heat transport processes (slowdown in the Atlantic meridional overturning circulation (AMOC) and increased heat transport out of the North Atlantic), but it is also linked to a poleward shift of

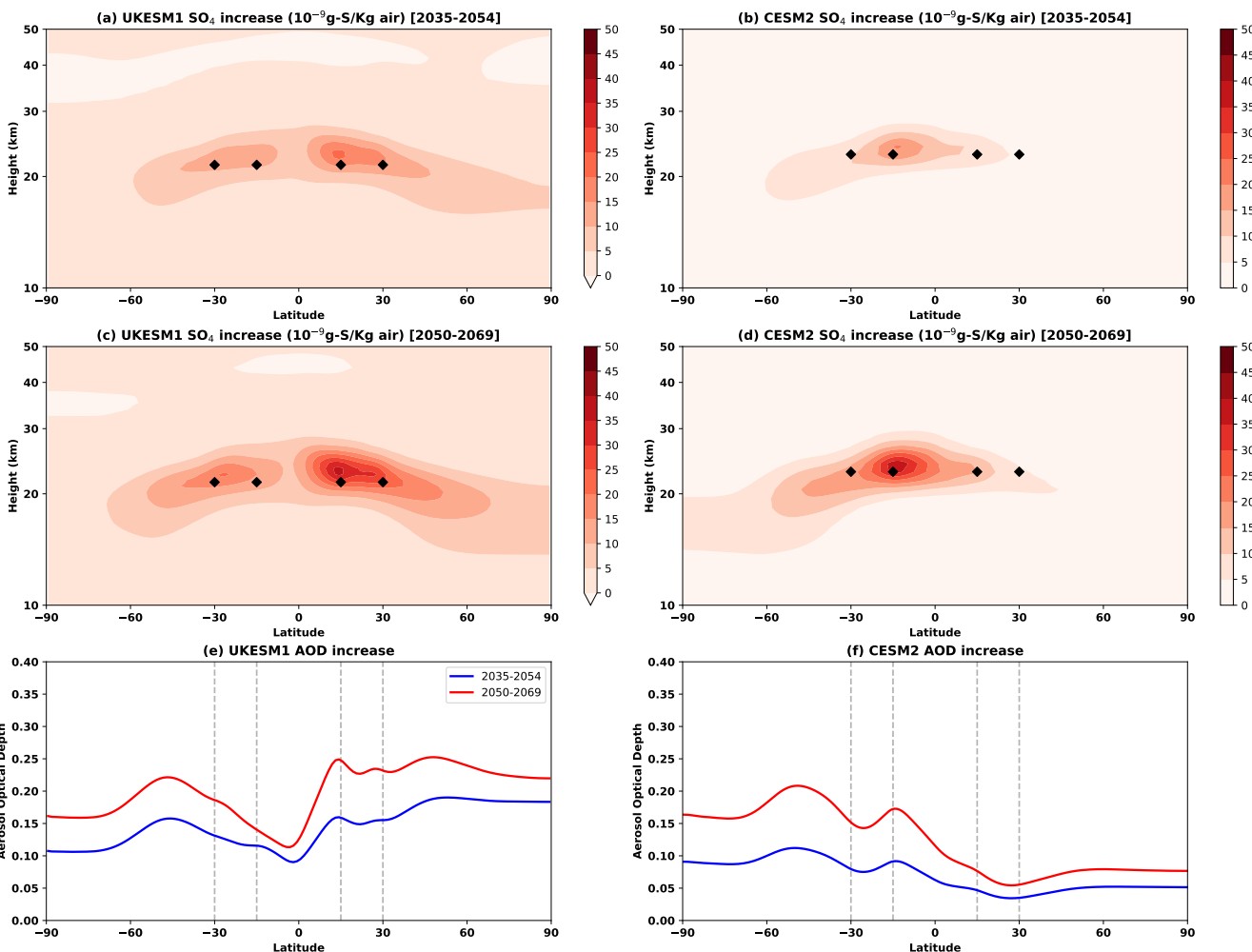

**Figure 3.** Comparison of zonal-mean ensemble-mean $SO_4$ increase in 2035-2054 (a,b) and 2050-2069 (c,d) between UKESM1 (left) and CESM2 (right), relative to their respective reference period mean of the SSP2-4.5 simulations. The black diamonds denote the injection locations. (e,f) Comparison of zonal-mean ensemble-mean aerosol optical depth (AOD) increase in 2035-2054 (blue) and 2050-2069 (red) between UKESM1 (left) and CESM2 (right), relative to their respective reference period mean of the SSP2-4.5 simulations.

westerly winds as a response to external forcings (Keil et al., 2020; He et al., 2022); and all CMIP6 models show a decline in AMOC in SSP2-4.5 including UKESM1 and CESM2 (Weijer et al., 2020). Figure A2 shows the AMOC for one ensemble member for UKESM1 and for the 10 ensemble members for CESM2 : both models show a decline in AMOC in SSP2-4.5 (≈25% for UKESM1 and ≈28% for CESM2), and a smaller decline in AMOC in ARISE-SAI-1.5 (≈18% for UKESM1 and ≈24% for CESM2). Thus it is unclear what drives the difference in North Atlantic temperature change between these two models, though this affects the pattern of temperature change outside of the area as the controller optimises for T0, T1, and T2.

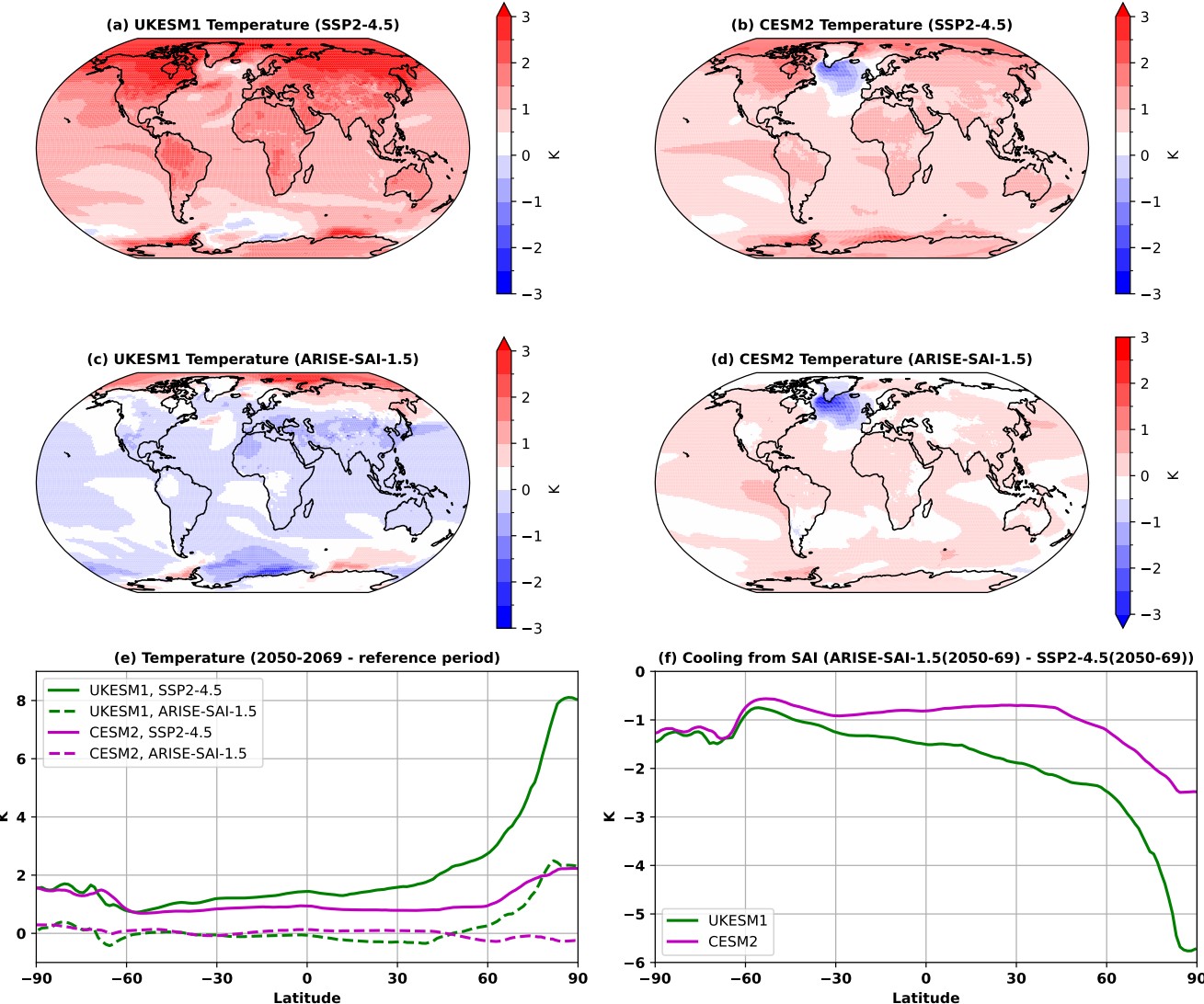

**Figure 4.** (a-d) Comparison of annual-mean ensemble-mean surface temperature change at 2050-2069 for UKESM1 and CESM2, and for SSP2-4.5 and ARISE-SAI-1.5, relative to their respective reference period mean. (e) Zonal-mean surface temperature change for UKESM1 and CESM2, and for SSP2-4.5 and ARISE-SAI-1.5. (f) The corresponding cooling from SAI.

Figure 5a and b show the time-series of global-mean precipitation for all simulations and figure 5c and d show the annual-mean temperature change as a function of the percentage change in annual-mean precipitation for both models and scenarios relative to their respective reference period for the global mean and the tropics only (between 30°N and 30°S). The effects of SAI on precipitation and other hydrological variables (reviewed in Ricke et al. (2023)) are very uncertain. The reduction in precipitation as a consequence of SAI is consistent with previous work (Irvine et al., 2019; Seeley et al., 2021), and is also

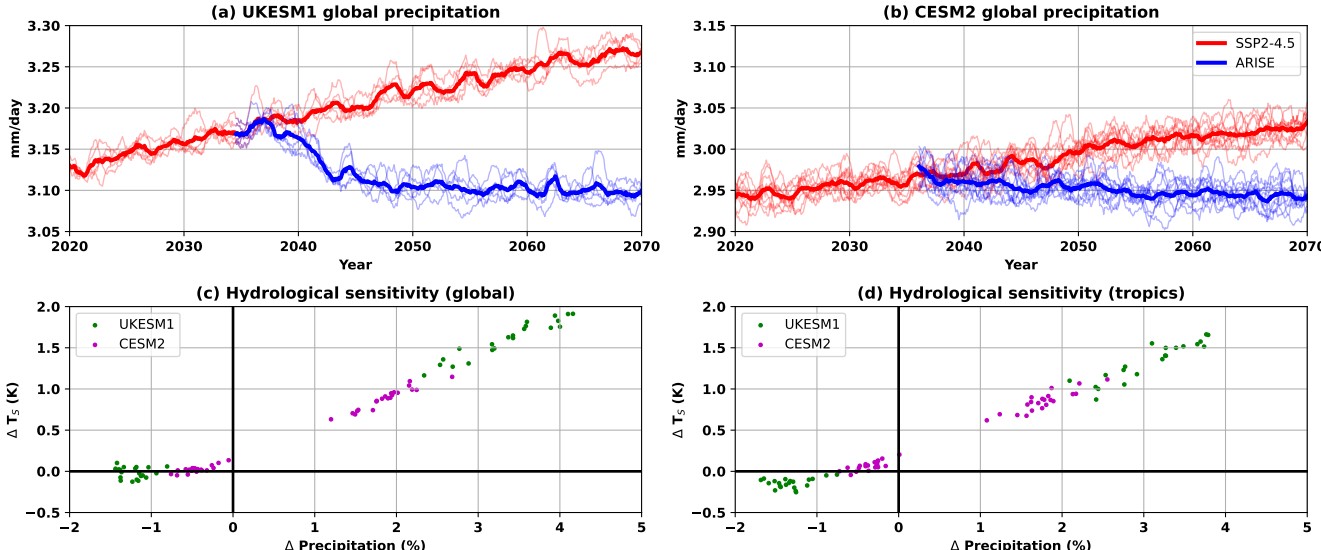

**Figure 5.** Comparison of global-mean precipitation for the SSP2-4.5 simulations (red) and the ARISE-SAI-1.5 simulations (blue) for UKESM1 (a) and CESM2 (b). Thin lines represent individual ensemble members, whereas the thick lines show the ensemble mean. (c) The annual-mean precipitation change as a function of temperature change for both models and both scenarios relative to their reference periods. (d) Same as (c) but for the tropics (between latitudes 30°N and 30°S).

consistent with observations of a spin-down in the hydrological cycle subsequent to the volcanic eruption of Pinatubo in 1991 (Trenberth and Dai, 2007).

The hydrological sensitivity is commonly defined as a change in precipitation for a given change in surface temperature. We find a good agreement in the hydrological sensitivity simulated under SSP2-4.5 for CESM2 and UKESM1. In contrast, in ARISE-SAI, the hydrological sensitivity is higher in UKESM1 than in CESM2: when the surface temperature is returned to that of the reference period, there is a ≈0.5% reduction in precipitation in CESM2 and a ≈1.2% reduction in UKESM1. This
apparent difference in the hydrological sensitivity may be due to the difference in the pattern of cooling. UKESM1 undercools the Arctic and overcools the tropics, and the change in global-mean precipitation is strongly influenced by tropical surface temperatures. Figure 5d reproduces the analysis of figure 5c focused on the tropics and shows a more consistent slope between temperature and precipitation changes between both models.

Figure 6 shows the seasonal precipitation change (December, January, and February (DJF) and June, July, and August (JJA)) for UKESM1 and CESM2 between 2050-2069 and their respective reference period, for SSP2-4.5 and ARISE-SAI-1.5. The annual-mean changes may hide important seasonal differences, for example in the monsoon precipitation, hence we choose to show seasonal changes here. The shaded areas indicate where the difference is not statistically significant, as evaluated using

a double-sided t-test with p < 0.05 considering all ensemble members and the 20 years as independent samples. We find large differences in the simulated precipitation responses to SAI between the two models. Importantly, large inter-model differences in the precipitation responses are also found for the SSP2-4.5 simulations without SAI. This is hardly surprising as it has long been known that climate models largely disagree on regional precipitation change (Box Ts.6 Figure 1 in Arias et al. (2021)). For example, in the Northern Hemisphere summer (JJA), there are important inter-model differences in key areas of high population and vulnerability such as Central Africa and South Asia. The uncertainty around precipitation changes in the climate with SAI is driven in part by the surface temperature change and in part by the direct effects of both increased greenhouse gases and stratospheric aerosols.

Figure 7 shows the zonal-mean change in land precipitation over land for DJF and JJA and for all simulations. The shaded area shows the ensemble standard deviation at each latitude point, whereas the thick lines show the ensemble mean. This lets us analyse the intra-ensemble variability in precipitation response and the difference in precipitation response between models and scenarios. Outside of the tropics, both models broadly agree for both DJF and JJA and for both scenarios. However, there are important differences in precipitation over tropical land (also seen in figure 6). In the reference SSP2-4.5 simulations, the precipitation response in JJA over land in the tropics are of opposite sign between UKESM1 and CESM2 (figure 7b). This is consistent with an increase in JJA T1 in the reference SSP2-4.5 simulations for UKESM1 and a decrease in JJA T1 for CESM2 (not shown), as the ITCZ generally migrates towards the warmer hemisphere. CESM2 has a larger standard deviation, which indicates a larger role for internal variability in its precipitation response. Finally, the precipitation changes over land under ARISE-SAI-1.5 are not larger than that under SSP2-4.5.

Figure 8 shows the different changes in projected extreme precipitation and reveals an interesting picture of the hydrological cycle under SAI. The change in the wettest pentad per year (annual maximum precipitation over 5 consecutive days; Tye et al. (2022)) is statistically insignificant over most land areas in UKESM1. The main exception is over Pakistan and northern India, where projected increases in summer mean rainfall (Figure 6g) may result from changes in the monsoon and correlate with projected increases in the wettest pentad. Projected decreases in dry spells (longest spell of consecutive days with <1mm rain per year) also correlate with regions of projected increases in UKESM1 seasonal mean precipitation (e.g. Sahel region, east Brazil). While many of the projected changes in both wettest pentad and dry spells are statistically insignificant, they do suggest a more muted hydrological cycle with fewer very heavy rain days and reduced persistence in wet and dry spells. As with mean precipitation, there are broad regions of agreement in the projected changes in wettest pentad and dry spells between UKESM1 and CESM2 at higher latitudes. However, CESM2 projects significant increases in the wettest pentad further south and east than UKESM1 with the greatest changes over the Horn of Africa and India/Bangladesh. The projected increases in CESM2 correspond to regions with changes in seasonal mean precipitation (Figure 6 f, h) and may be related to a change in the simulated location of the Inter Tropical Convergence Zone and monsoon systems. Such changes are consistent with the projected increase in mean precipitation over the Northern Indian Ocean and India, and decrease south of the Equator. Differences between UKESM1 and CESM2 are also unsurprising given the known disagreements between regional precipitation changes.

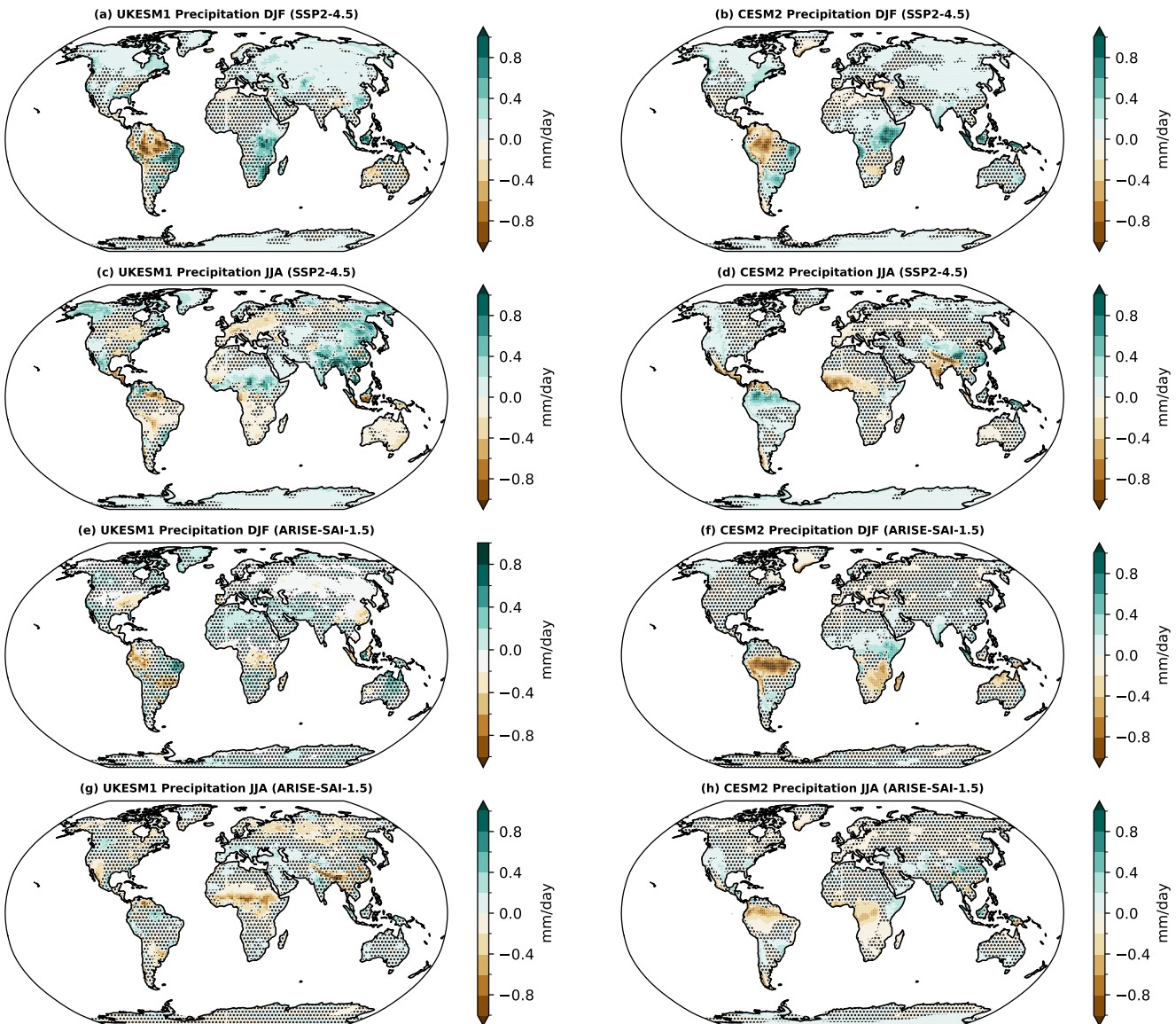

**Figure 6.** Comparison of ensemble-mean seasonal precipitation change at 2050-2069 for UKESM1 and CESM2, for SSP2-4.5 and ARISE-SAI-1.5, and for DJF and JJA, relative to their respective reference period mean. DJF refers to December, January, and February, and JJA refers to June, July, and August. Shaded areas indicate where the difference is not statistically significant, as evaluated using a double-sided t-test with p < 0.05 considering all ensemble members and 20 years as independent samples.

Projected decreases over the Amazon and Central Africa in CESM2 appear to be associated with changes in the persistence of
wet and dry days, with significant decreases in the longest spells of wet days (not shown) and increases in dry spells. While
the patterns of changes in extreme precipitation are also very different for UKESM1 and CESM2 under SSP2-4.5, their main

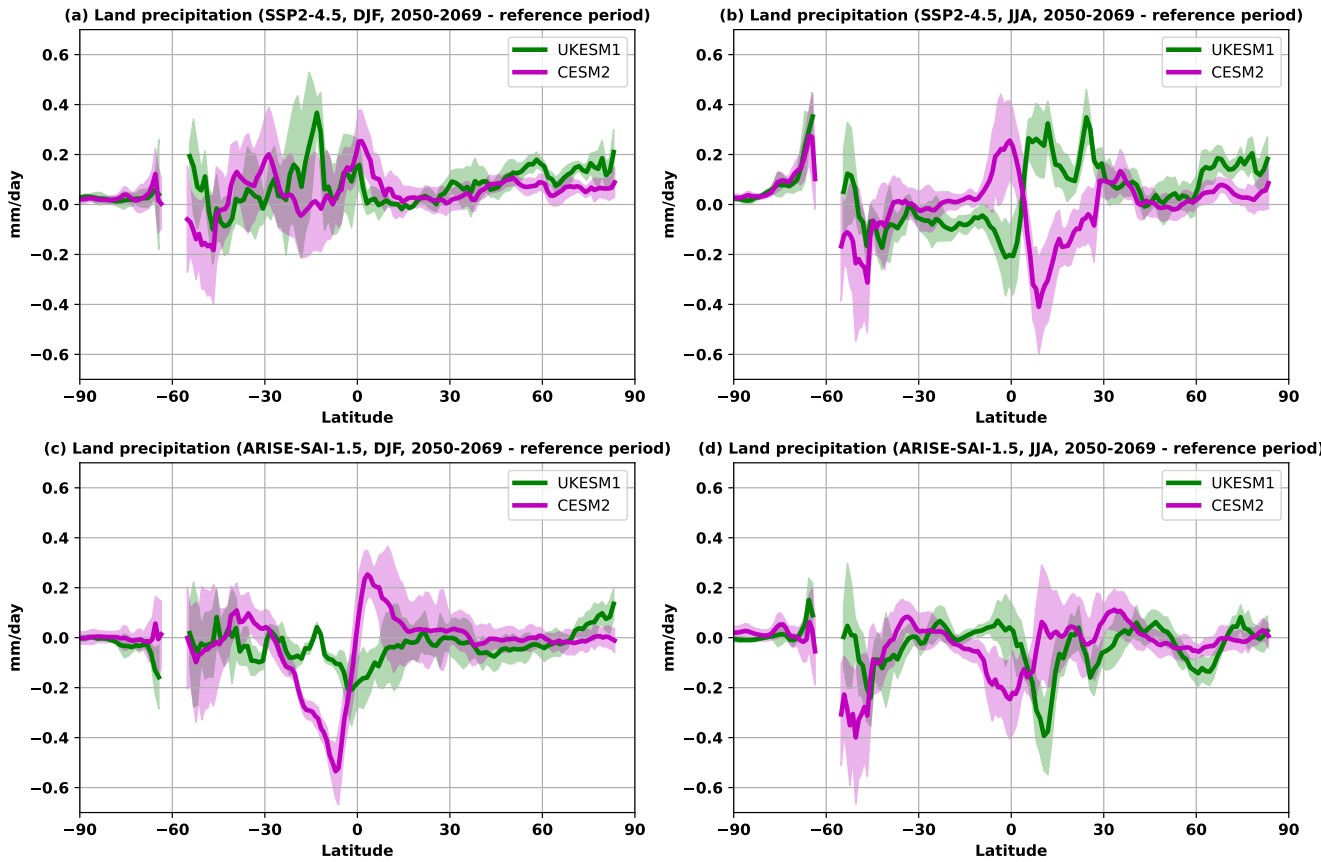

**Figure 7.** Zonal-mean land precipitation change for SSP2-4.5 (a,b) and ARISE-SAI-1.5 (c,d) for DJF (a,c) and JJA (b,d). The shaded area shows the standard deviation at each latitude point, whereas the thick lines show the ensemble mean. DJF refers to December, January, and February, and JJA refers to June, July, and August.

similarity is that extreme precipitation is projected to increase in frequency and intensity everywhere at varying levels regardless of whether the region will otherwise be wetter or drier.

## 4 Conclusions

We have described two ensembles of simulations of stratospheric aerosol injection (SAI) using CESM2 and UKESM1, which are part of a set entitled "Assessing Responses and Impacts of Solar climate intervention on the Earth system with Stratospheric Aerosol Injection (ARISE-SAI)". In this ensemble, we begin SAI in 2035 with the target of maintaining global-mean surface temperatures at 1.5 K above pre-industrial levels, hence it is called ARISE-SAI-1.5. This set of simulations seeks to increase our understanding of the impacts of climate interventions using stratospheric aerosols in a policy-relevant scenario. The first

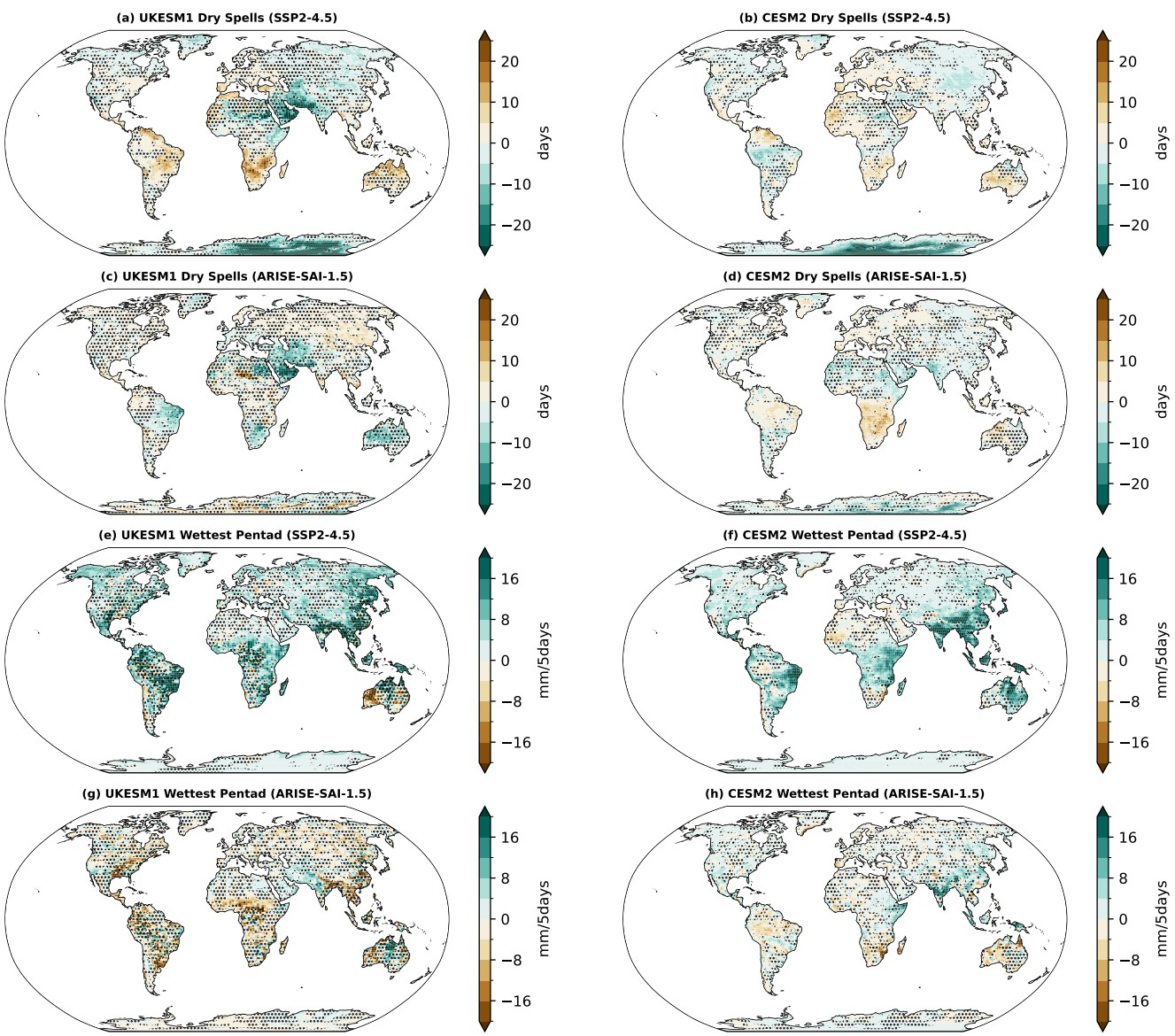

**Figure 8.** Comparison of ensemble-mean change in dry spells (longest consecutive spell of days with <1mm rain per year) and wettest pentad (annual maximum precipitation over 5 consecutive days) at 2050-2069 for UKESM1 and CESM2, and for SSP2-4.5 and ARISE-SAI-1.5, relative to their respective reference period mean. Shaded areas indicate where the difference is not statistically significant, as evaluated using a double-sided t-test with p < 0.05 considering all ensemble members and 20 years as independent samples.

ensemble of simulations using CESM2 was described in Richter et al. (2022). This is both the first implementation of a multi-latitude strategy using a control algorithm outside of CESM and the first model intercomparison of such strategies.

The key elements of the model comparison are summarised below:

– There is a general lack of consistency between the models in the resultant locations of the injection strategy with CESM2 injecting the greatest fraction of sulfur dioxide at 15°S, while UKESM1 injects at 30°N and 30°S before swapping to 15°N later on in the simulations. This is in part caused by the pattern of forcing from the increased stratospheric aerosols which is unable to perfectly counter the greenhouse gas forcing and the climate model's internal feedbacks which, for UKESM1, favours a strong Arctic amplification of surface temperature change. This emphasises the need for research

which further constrains the level of future Arctic warming, as it will inform any hypothetical future SAI deployment strategy which seeks to maintain interhemispheric and equator-to-pole temperature differences. Other factors such as the cloud adjustment to $CO_2$, the North Atlantic warming hole, and effects of tropospheric aerosol changes also play a role in the differences in injection strategy (Fasullo and Richter, 2022).

    – Both models have substantially less surface warming in the ARISE-SAI-1.5 simulations compared to SSP2-4.5, with

a strong North Atlantic warming hole for CESM2. CESM2 is successful in reaching its temperature targets, though the interhemispheric temperature difference and the equator-to-pole temperature gradient do not change much in the reference SSP2-4.5 scenario. In contrast, UKESM1 has a very strong Arctic amplification of warming (a ratio of 3.6 in SSP2-4.5 compared to 2.1 for CESM2 when defined as warming northward of 70°N relative to the global mean).

    – Outside the tropics, both models agree on precipitation changes over land for both summer and winter and both scenarios.

The changes over tropical land though are more complex. In the reference SSP2-4.5 simulations, the precipitation response over land in the tropics are of opposite sign between UKESM1 and CESM2 (figure 8b). For CESM2, the standard deviation in the precipitation response over tropical land is larger, suggesting an important role for internal variability. Finally, the precipitation changes over land under ARISE-SAI-1.5 are not larger than that under SSP2-4.5.

This ensemble comparison is the first step in comparing more policy-relevant scenarios of SAI by forcing two Earth System

Models to achieve a model-specific set of global surface temperature targets by injecting $SO_2$ at multiple predefined locations. This constraint on the surface temperature pattern forces the injection amounts at each location, the AOD pattern, and the forcing from stratospheric aerosols to be different. Thus, although CESM2 succeeds in achieving the T0, T1, and T2 targets, UKESM1 only partially achieves the T2 target.

As shown by Zhang et al. (2023), while the controller minimises residual changes in the climate, a hemispherically symmetric strategy with injection off the equator and controlling only for the global-mean surface temperature strikes a good balance between ease of implementation and minimising residual climate change. Multi-model simulations (e.g. in the context of GeoMIP) need to balance the complexity of implementation of SAI strategies, with pragmatic considerations about the ease of implementation to maximise participation and enable statistically robust conclusions to be drawn (e.g. Visioni et al., 2023b).

Thus, it is suggested that future model intercomparisons of SAI, where model simulations would use nominally identical injection strategies, will likely follow this simplified protocol instead of the more complex control algorithm shown here.

Unlike CESM2, in UKESM1, injecting at 15°N and injecting at 30°N yield a very similar pattern of surface temperature change, thus the controller-defined injection strategy varies significantly for small changes in surface temperature, which explains the large increase in injection at 15°N after 2055 in these simulations. This implies that, in UKESM1 and other climate models, the injection location may need to be adjusted to yield distinct enough temperature change patterns. And, if aircraft were to be used for deployment, any practical logistics would need to be as efficient as possible given the quantities of $SO_2$ required and the limitations and trade-offs between aircraft payload and fuel capacity (e.g. Smith, 2020). The current uncertainty around injection strategies in the models suggests that, if our objective is minimising changes in the large-scale pattern of temperature change, we cannot anticipate the required infrastructure required for real-world SAI delivery. It is also worth noting that injection strategies will be strongly influenced by the pattern of warming in future climate change scenarios, which differ markedly between UKESM1 and CESM2. This provides additional motivation to better understand and to validate the patterns of temperature response in the absence of SAI. Hence, along with more models participating in a simpler future model intercomparison as described above, a much better understanding of the fidelity of model performance would also seem a prerequisite for practical deployment.

*Code and data availability.* The code to reproduce the figures is available at https://github.com/matthewjhenry/arise_comparison_acp. The data is available at https://zenodo.org/record/6473954 for the CESM2(WACCM6) SSP2-4.5 simulations and at https://zenodo.org/record/6473775 for the CESM2 ARISE-SAI-1.5 simulations; extreme precipitation indices are available at https://zenodo.org/record/7552583 for the CESM2 simulations and at https://zenodo.org/record/7922503 for the UKESM1 simulations. Complete output from all 10 members of CESM2(WACCM6) SSP2-4 simulations and ARISE-SAI-1.5 simulations is freely available the NCAR Climate Data Gateway at https://doi.org/10.26024/0cs0-ev98 and https://doi.org/10.5065/9kcn-9y79 respectively. Data for the UKESM1 ARISE-SAI-1.5 simulations is available at https://data.ceda.ac.uk/badc/dep and the data for the UKESM1 SSP2-4.5 simulations is available on the Earth System Grid Federation database. We anticipate community analysis of various aspects of the Earth system of the ARISE-SAI-1.5 simulations. There is no obligation to inform the project leads about the analyses you are performing, but it would be helpful in order to coordinate analysis and avoid duplicate efforts.

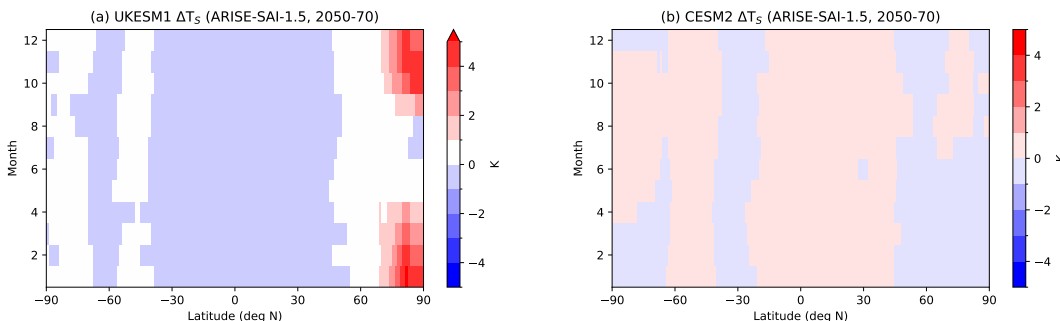

**Figure A1.** Seasonality of zonal-mean surface temperature change at 2050-70 for ARISE-SAI-1.5, for UKESM1 (a) and CESM2 (b), relative to each model's reference period.

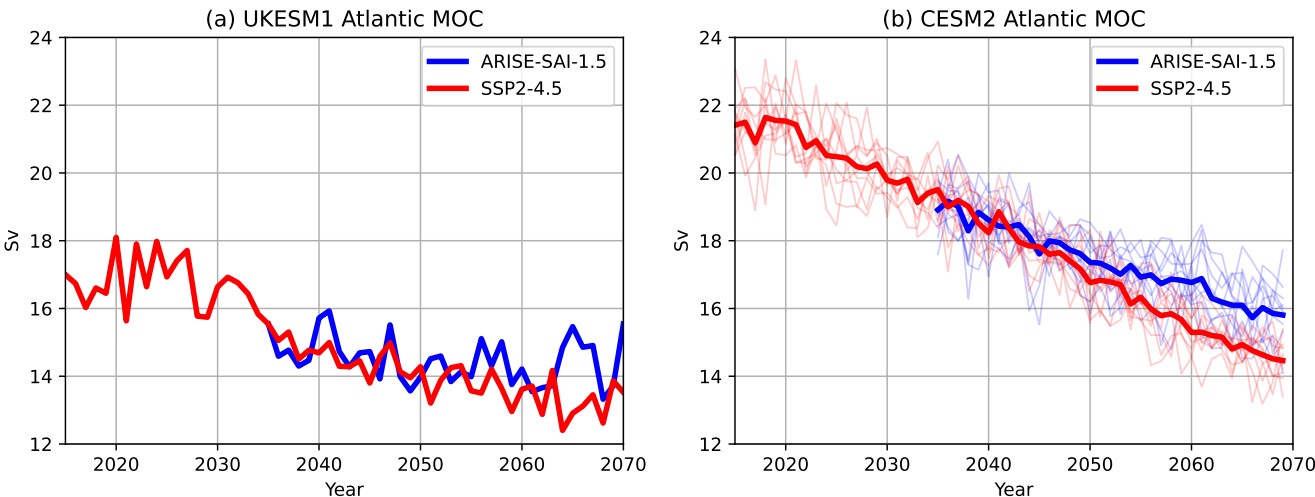

**Figure A2.** Atlantic Meridional Overturning Circulation for UKESM1 and CESM2 in Sverdrup (Sv). Calculated as the maximum across depth and latitude of the ocean circulation streamfunction in the Atlantic basin. UKESM1 only has data for one ensemble member, and for CESM2 the thick lines show the ensemble-mean and the thin lines show each ensemble member.

*Author contributions.* MH and JH wrote the manuscript with contributions from all co-authors. MH made the figures with contributions from MT. DV, WRL, and DM designed and implemented the controller algorithm in CESM2. MD adapted the controller algorithm to UKESM1 and AJ conducted the ensemble of simulations. MT contributed the figure and section on climate extremes.

*Competing interests.* The authors declare they have no conflict of interest.

*Acknowledgements.* MH is funded by the Natural Environment Research Council Exeter-NCAR (EXTEND) collaborative development grant (NE/W003880/1) and by SilverLining through the Safe Climate Research Initiative. JMH, AJ, and MD were supported by the Met Office Hadley Centre Climate Programme funded by the Department for Business, Energy, and Industrial Strategy (UK) and by Silverlining through its Safe Climate Research Initiative. MT was supported by Silverlining through its Safe Climate Research Initiative and by the National Center for Atmospheric Research, which is a major facility sponsored by the National Science Foundation under Cooperative Agreement No. 1852977. DV, EMB, DGM and WL acknowledge support from the Atkinson Center for a Sustainability at Cornell University and from the National Science Foundation through agreement CBET-2038246. EMB also acknowledges support from the NOAA cooperative agreement NA22OAR4320151 and NOAA Earth's Radiative Budget initiative. For the purpose of open access, the author has applied a Creative Commons Attribution (CC BY) licence to any Author Accepted Manuscript version arising.

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
