# Peer review of "Comparison of UKESM1 and CESM2 Simulations Using the Same Multi-Target Stratospheric Aerosol Injection Strategy"

_EGUsphere, 2023_

## Author Comment (AC2)

**Response to reviewers**

We thank both reviewers for their helpful comments. Reviewer 2 is broadly supportive, and reviewer 1 made some important points about the need to go beyond stating the differences between models. We had already included quite some discussion in the manuscript about the significant differences in the baseline simulations (SSP2-4.5) between the models, and how this changes with reference to the T0, T1, and T2 targets. However, we have added a better explanation for the change in injection strategy for UKESM1 and made some changes to the manuscript to reflect on the broader implications of this analysis and future model comparisons of SAI. The focus of the study is to perform an initial analysis documenting i) that we have implemented a reasonably sophisticated controller in UKESM1 to determine annual injection rates as had been done for CESM2, ii) that the controller 'works' to some extent i.e. that the T0, T1 targets are able to be maintained but the T2 target, while improved compared to the baseline global warming scenario, is less well satisfied in UKESM1, iii) that in contrast to CESM2, the temperature response in UKESM1 to injections at 15N and 30N is quite similar, thus small changes in T1 will result in large changes in the controller-defined distribution of injection (which happens after 2055), iv) implications for designing controllers in other climate models.

We believe that these results are important enough to communicate to the scientific community.

A thorough understanding of why the two models respond differently to SAI is beyond the scope of this work. It would, for example, require understanding why the two models have a different climate sensitivity, which is a field of research of its own. However, this manuscript can provide a starting point for further analysis of the differences in the climate response to the same multi-target SAI strategy, and for understanding which uncertainties matter most when considering different climate models in the context of SAI.

Please find below their **comments** and our responses.

**Reviewer 1**

*I'm having a difficult time reviewing this paper because I'm not entirely sure what I should say. It is the first paper to compare two models using multi-latitude control in SAI, which is a great idea. But a lot of the analysis, results, and conclusions are basically a form of "different things are different". I don't know what the research community is supposed to do with that. The authors say at the end that more models should do this experiment, but that doesn't seem like a useful effort if the best we'll be able to do is that we'd just get even more different results.*

We agree that this manuscript only shows results from a particular experimental design, and further work, beyond the scope of the current paper, will be required to fully understand the reasons for these differences. We need to restrict the scope of the manuscript to a manageable chunk of work and hope it can provide a starting point for further analysis of the differences in the climate response to the same multi-target SAI strategy.

As noted above, we add a better explanation for the difference in the controller behaviour between both models and implications for designing controllers for other climate models :

"In UKESM1, the temperature response to injection at 15N only and 30N only is quite similar, in contrast with CESM2 (Visioni et al. 2023). Thus the influence matrix between injection rates and

changes in T0, T1, and T2 is almost singular, therefore small changes in what the controller needs to respond to can result in large changes in the controller-defined injection rates, as observed after 2055 in these simulations. In an effort to correct for an increasing error in meeting the T1 objective caused by continued Arctic warming and slight southern hemisphere cooling after 2050, the controller increases the fraction of injection at 15N. However, redistributing the injection rates is unable to significantly change T1, resulting in further increases in the fraction injection at 15N. Therefore the set of injection latitudes chosen in CESM2 during initial studies to simultaneously meet these temperature objectives (MacMartin et al. 2017) might not necessarily work in other climate models and may need to be adjusted to ensure that the set of injection latitudes yield distinct influences on the zonal-mean temperature response."

***There is a goldmine of information in the simulations the authors analyze. I would encourage them to dig into mechanisms and provide some insights. If we know why the models are giving us different answers, that can tell us about what the most important uncertainties might be.***

***As an example, the authors say on line 136 that the three factors are expected to be different in UKESM1. So why not look at them?***

We refine the sentence as follows : "The fast cloud-adjustment to $CO_2$ is expected to be different in other climate models (Smith et al. 2020, Wang et al. 2021). However as shown in Fasullo and Richter (2022), separating the fast adjustment from the surface temperature dependent response requires further idealized experiments, which are beyond the scope of the present study. The baseline scenario (SSP2-4.5) is the same for both models; nonetheless, the response to the same aerosol forcing and short-lived greenhouse gases forcing might be different between the two models (Smith et al. 2020). The North Atlantic warming hole, however, is not present in UKESM1 (figure 4) and is expected to be different for other climate models (further discussed in section 3.4). These differences result in a difference in the distribution of injection rates that are required to satisfy the T1 objective."

***On line 210 you talk about the hydrologic sensitivities being similar or different depending on the context/simulation. So what are they, and why are they different?***

We have made several changes to the hydrological sensitivity section in response to reviewer 2, we hope this makes things clearer.

***When you say "likely" on line 241, can you do better?***

We have added a qualifier to the sentence related to the simulated changes in the location of the Inter Tropical Convergence Zone. And an additional explanatory sentence: "Such changes are consistent with the projected increase in mean precipitation over the northern Indian Ocean and India, and decrease south of the Equator. Differences between UKESM1 and CESM2 are also unsurprising given the known disagreements between regional precipitation changes."

**I'm not listing every example in the paper, but hopefully the authors see what I mean.**

**I also think the authors are missing an opportunity with the multi-model comparison to talk about what this study could mean. If multiple plausible models are getting a different range of achievable climates, that's really useful to know in terms of the limitations of SAI (or at least SAI with a four latitude strategy).**

We agree that it is useful to clarify the broader meaning of this work, so we edit the conclusions as follows:

"This ensemble comparison is the first step in comparing more policy-relevant scenarios of SAI by forcing two Earth System Models to achieve a model-specific set of global surface temperature targets by injecting SO2 at multiple predefined locations. This constraint on the surface temperature pattern forces the injection amounts at each location, the AOD pattern, and the forcing from stratospheric aerosols to be different. Thus, although CESM2 succeeds in achieving the T0, T1, and T2 targets, UKESM1 only partially achieves the T2 target.

As shown by Zhang et al. 2023, while the controller minimises residual changes in the climate, a hemispherically symmetric strategy with injection off the equator and controlling only for the global-mean surface temperature strikes a good balance between ease of implementation and minimising residual climate change. Multi-model simulations (e.g. in the context of GeoMIP) need to balance the complexity of implementation of SAI strategies, with pragmatic considerations about the ease of implementation to maximise participation and enable statistically robust conclusions to be drawn (e.g. Visioni et al. (2023b)). Thus, it is suggested that future model intercomparisons of SAI, where model simulations would use nominally identical injection strategies, will likely follow this simplified protocol instead of the more complex control algorithm shown here.

Unlike CESM2, in UKESM1, injecting at 15N and injecting at 30N yield a very similar pattern of surface temperature change, thus the controller-defined injection strategy varies significantly for small changes in surface temperature, which explains the large increase in injection at 15N after 2055 in these simulations. This implies that, in UKESM1 and other climate models, the injection location may need to be adjusted to yield distinct enough temperature change patterns. And, if aircraft were to be used for deployment, any practical logistics would need to be as efficient as possible given the quantities of SO2 required and the limitations and trade-offs between aircraft payload and fuel capacity (e.g. Smith (2020)). The current uncertainty around injection strategies in the models suggests that, if our objective is minimising changes in the large-scale pattern of temperature change, we cannot anticipate the required infrastructure required for real-world SAI delivery. It is also worth noting that injection strategies will be strongly influenced by the pattern of warming in future climate change scenarios, which differ markedly between UKESM1 and CESM2. This provides additional motivation to better understand and to validate the patterns of temperature response in the absence of SAI. Hence, along with more models participating in a simpler future model intercomparison as described above, a much better understanding of the fidelity of model performance would also seem a prerequisite for practical deployment."

**I noticed there is a lot of emphasis on Arctic amplification, especially differences in Arctic amplification between the two models. But SAI does a pretty good job of reducing Arctic warming in both models. So I have to wonder how important model differences in AA actually are. If you did some quantification of feedbacks or mechanisms, we might be able to figure out how important AA is.**

AA is important in explaining the reasons for the different injection distributions, but the reviewer is right in that the disagreement between models in regional surface temperature change is much reduced in the geoengineered climates, as the latitudinal pattern in cooling is similar to the latitudinal pattern in warming. However, the coupled nature of the Arctic climate makes it difficult to quantify the role of different feedbacks in producing Arctic Amplification. Again, such an analysis is beyond the scope of this work and further investigation is required in this area.

We add the following:

"The coupled nature of the Arctic climate makes it difficult to quantify the role of individual mechanisms in the Arctic amplified warming, which were reviewed in Previdi et al. (2021) and Taylor

et al. (2022). The uncertainty in the sea ice feedback would be reduced in a world with SAI as the sea ice would be restored. However, the uncertainties in cloud response to CO2 and aerosol-cloud interactions would be at least as important."

**A few more comments:**

**Figure 1c: It looks like you have two different timescales of response here, in that there's a clear shift around 2045. Kravitz et al. (2016) found something similar, which may suggest some mechanisms that you could pursue to explain this behavior. Relatedly, I acknowledge that the controller is prioritizing T1 over T2, so I would expect deviations in T2 from the reference value. But why is T1 not controlled very well?**

This is a good point and is related to our new discussion of the behaviour of the controller in UKESM1 (first response to reviewer 1). As described in our explanation, the change in T1 is caused by continued warming of the Arctic and slight cooling of the Southern hemisphere after 2050.

**Line 26: some missing words**

Fixed, thank you.

**Section 2: Be more specific about what the temperature targets are and the control gains.**

Fixed, thank you.

**Reviewer 2**

We thank the reviewer for their helpful comments and feedback. Please find below their **comments** and our responses.

*This paper makes the first comparison of two models implementing the same feedback control style implementation of SAI, and as such it makes a major contribution to the literature. The study compares the realized temperatures against the target temperatures for the models, finding that UKESM1 misses these targets due to substantial arctic amplification, and compares the injection strategies needed and resultant aerosol burdens, noting substantial differences, including a large southern-hemisphere bias in CESM2. It also compares the hydrological response of the two models, finding substantial differences between the models and significant shifts in tropical precipitation in both models. There is some useful discussion of the potential reasons behind the differences in the aerosol distribution and failure of UKESM1 to achieve the temperature target, though there could have been more discussion of the implications of the inter-model differences in the experiment-as-realized for the design of future feedback control experiments intended for intermodel comparisons.*

*I'd recommend the article for publication after the following, fairly minor comments are addressed.*

*The introduction gives a fine introduction to the topic, the history of the field and of modelling efforts. However, it doesn't devote much text at all to reviewing the climate response seen in previous studies or to setting up theoretical expectations about what we may expect from the comparison. Would be nice to see a bit more of this up front.*

We had mentioned that injecting at multiple locations with a controller ameliorates the side effects of tropical injections. We add the following (after "started in 2020" L.52 in 1st version) : "These simulations show that the interhemispheric and equator-to-pole temperature targets can be met, even in a high greenhouse gas emission scenario in CESM1. However, they also show that the hydrological cycle is suppressed relative to the target climate, results that appear common to many other SAI strategies (e.g. Tilmes et al., 2013; Irvine et al., 2019)."

*The results (and discussion) section generally does a good job of explaining the findings and putting them in the context of previous results, but there could be more of this for the hydrological results. This could be quite simple in that climate models are notorious for disagreeing on regional hydrological change, and so noting that the fact that these models disagree here is hardly surprising.*

We add "This is hardly surprising as it has long been known that climate models largely disagree on regional precipitation change (Box Ts.6 Figure 1 in \cite{IPCC_AR6_WG1_TS} for example)." in the Figure 6 paragraph.

*The conclusion section could benefit from some discussion of the fact that while the models followed the same experimental procedure and so nominally are simulating the same experiment, they are in effect quite different experiments-as-realized. That is, they produce different patterns of temperature change, driven by different patterns of forcing (and they have a different baseline, though this seems less significant). To what extent are the authors comparing like with like? What are the implications of this? What changes in setup could produce experimental designs that would be more suitable for a future model intercomparison*

***or are such profound differences in experiment-as-realized inevitable? If so, what does this imply for the assessment of SAI?***

These are great questions and we thank the reviewer for them.

We modify the last paragraph of the conclusion as follows:

"This ensemble comparison is the first step in comparing more policy-relevant scenarios of SAI by forcing two Earth System Models to achieve a model-specific set of global surface temperature targets by injecting SO2 at multiple predefined locations. This constraint on the surface temperature pattern forces the injection amounts at each location, the AOD pattern, and the forcing from stratospheric aerosols to be different. Thus, although CESM2 succeeds in achieving the T0, T1, and T2 targets, UKESM1 only partially achieves the T2 target.

As shown by Zhang et al. 2023, while the controller minimises residual changes in the climate, a hemispherically symmetric strategy with injection off the equator and controlling only for the global-mean surface temperature strikes a good balance between ease of implementation and minimising residual climate change. Multi-model simulations (e.g. in the context of GeoMIP) need to balance the complexity of implementation of SAI strategies, with pragmatic considerations about the ease of implementation to maximise participation and enable statistically robust conclusions to be drawn (e.g. Visioni et al. (2023b)). Thus, it is suggested that future model intercomparisons of SAI, where model simulations would use nominally identical injection strategies, will likely follow this simplified protocol instead of the more complex control algorithm shown here.

Unlike CESM2, in UKESM1, injecting at 15N and injecting at 30N yield a very similar pattern of surface temperature change, thus the controller-defined injection strategy varies significantly for small changes in surface temperature, which explains the large increase in injection at 15N after 2055 in these simulations. This implies that, in UKESM1 and other climate models, the injection location may need to be adjusted to yield distinct enough temperature change patterns. And, if aircraft were to be used for deployment, any practical logistics would need to be as efficient as possible given the quantities of SO2 required and the limitations and trade-offs between aircraft payload and fuel capacity (e.g. Smith (2020)). The current uncertainty around injection strategies in the models suggests that, if our objective is minimising changes in the large-scale pattern of temperature change, we cannot anticipate the required infrastructure required for real-world SAI delivery. It is also worth noting that injection strategies will be strongly influenced by the pattern of warming in future climate change scenarios, which differ markedly between UKESM1 and CESM2. This provides additional motivation to better understand and to validate the patterns of temperature response in the absence of SAI. Hence, along with more models participating in a simpler future model intercomparison as described above, a much better understanding of the fidelity of model performance would also seem a prerequisite for practical deployment."

***Specific comments***

***L30 – I'd suggest sticking to one of SRM or solar geoengineering or solar climate intervention and using throughout.***

This is fixed, we refer mainly to SAI.

***L40 – non-synergistic – another term?***

Changed to uncoordinated.

**L45 – "consistently" instead of commonly?**

Changed to consistently.

**L56-60 – "ensemble" seems a bit grand to describe a pair of models.**

Changed to comparison.

**L66 – spare parentheses?**

No, CESM2(WACCM6) is the full name of the model.

**L63-79 – Would be good to mention the ocean models too.**

Done.

**L93 – what longitude? into a single gridcell?**

Fixed, 180 for both models.

**L101 – if these definitions are short, just include them here or describe the equation.**

T1 and T2 are defined 2 sentences prior in the parentheses.

**L103-110 – It would be good to mention what CESM2's temperature increase relative to its preindustrial is. And to have some discussion of the consequences of this appear at an appropriate point in the paper.**

We adjust this paragraph accordingly to include these numbers and the discussion.

**L112 - "Results and discussion" would be more accurate.**

Fixed.

**L125-155 – I found this difficult to follow. It would be clearer to present the figure first and then compare that to other studies, or to not mention the figure until you wish to describe it. Also in describing the results of Richter et al. 2022 are you also describing Figure 2b?**

Agreed, we move the "A comparison of injection rates between UKESM1 and CESM2 chosen by the controller is given in Figure 2." sentence to the beginning of the following paragraph.

**Figure 2 – Hot / Cold colours or similar to indicate north vs. south would make it clearer that Arise is doing something surprising in putting so much in one hemisphere. Some text to this effect would also be good, e.g., Reporting the percentage in NH vs SH would make this more obvious to the reader rather than just stating that it "required more" in the SH.**

Agreed, we make the suggested changes to the figure.

And great point about the percentages, we change the text as follows : "In CESM2, 82% of the injection occurs in the Southern hemisphere by 2070, whereas in UKESM1 it is only 27%."

**L165 – perhaps make the change of focus clearer, by saying: "this reduced lifetime is compensated by the injection being 50% greater" or something to that effect.**

We make the suggested change.

*L163-171 – Here as well a quantitative comparison of NH vs SH burden or AOD would be nice, e.g. in "CESM2 75% of the SO4 is in the SH"*

Agreed, we add that for CESM2, the NH change in AOD is 31% of the total, whereas for UKESM1, it is 57% of the total.

*L169 – is this comparison backwards? Doesn't the SO4 cause the AOD increase?*

This is the way it is expressed in Visioni et al. 2023, i.e. as a SO4 increase per 0.1 AOD. This comparison allows us to understand how much burden alone increases for a unit increase in AOD, which both compounds information on burden and size and is a commonly used measure.

*L170 - "not shown" or provide some citation.*

Good point, we add a reference to Visioni et al. 2023.

*Figure 4, 6, and 7 – These map plots all have overly long titles (cut dates?) and too much white space. The hydrological maps are hard to read. Try to make these map plots as large as possible.*

Agreed, we make these changes.

Hydrological changes are not as spatially consistent as temperature changes, which make the plots hard to read when statistical significance hatching is included.

*L196 – and a smaller decrease in Arise. A recovery suggests time has passed.*

Fixed.

*L197 – Wouldn't this blob have a greater effect on T1 and T2 than global?*

True the blob will affect all other regions to some extent as the controller optimizes for T0, T1, and T2. This sentence is edited as follows : "Thus it is unclear what drives the difference in North Atlantic temperature change between these two models, though this affects the pattern of temperature change outside of the area as the controller optimises for T0, T1, and T2."

*L209-216 – I'd suggest considering reporting results in %/C as this is how I understand hydrological sensitivity and as the models show quite different temperature responses. I'd also need more convincing that the pattern of temperature is the main driver, rather than the substantially different magnitude of cooling. Perhaps some comparison to the literature would help make the case.*

We have modified the figure to be plotted with % change in precip in x-axis and temperature change in y-axis, as in Tilmes et al. 2013. This way of plotting it makes it easier to put all simulations in one figure. Hence we bring the appendix figure as a new panel to the current figure.

We find that this better illustrates our point that it is the overcooling of the tropics in UKESM1 which causes a larger reduction in precipitation globally, thus the hydrological sensitivity is not so different between models. We have changed the figure caption and text accordingly.

*Figure 5 – I'm presuming 1.5C here means "the baseline" Which is defined differently in the two models and doesn't correspond to the model's version of 1.5C in the case of CESM2. I'd suggest switching to just temperature anomaly.*

Good point. We changed it to temperature anomaly, the figure caption already refers to the reference period.

*L218-225 – As previous sections cited existing literature it's probably worth noting that regional differences in precipitation between models is common.*

We add "This is hardly surprising as it has long been known that climate models largely disagree on regional precipitation change (Box Ts.6 Figure 1 in \cite{IPCC_AR6_WG1_TS} for example)." in "Figure 6" paragraph.

*L226-227 – Not sure exactly what this last line means, e.g., the uncertainty in the precip response due to GHGs includes uncertainty in its effect on temperature, whereas here that temperature-based uncertainty is eliminated. The fast, forcing-driven effects are combined in Arise, but the slow-temperature driven ones are eliminated.*

Yes, maybe "compound product" is inaccurate. We replace with "The uncertainty around precipitation changes in the climate with SAI is driven in part by the surface temperature change and in part by the direct effects of both increased greenhouse gases and stratospheric aerosols."

*L241-243 – How is the causality being determined here?*

Agreed that it's hard to draw a causal relationship here. Determining the causal chain is beyond the scope of this manuscript, so we change the wording to "appear to be associated with changes".

*Figure 8 – Would seem more useful for this to appear directly before or after the figure on mean precipitation.*

Agreed, we swap figures 7 and 8.

*L256 – perhaps clarify what exacerbation means in this context.*

Changed to "Finally, the precipitation changes over land under ARISE-SAI-1.5 are not larger than that under SSP2-4.5."

*L265-end – Given the length of these bullet points, I'd suggest reframing as normal prose.*

I am inclined to keep them as bullet points, especially as the page layout makes it so it does not take so much extra space.

*L274-275 – This isn't particularly clear.*

We have changed the explanation around the injection strategy in UKESM1.

*L281-289 – Seems odd to have the injection last.*

Agreed, we move it to the first bullet point.